# Gut Microbiome Changes Occurring with Norovirus Infection and Recovery in Infants Enrolled in a Longitudinal Birth Cohort in Leon, Nicaragua

**DOI:** 10.3390/v14071395

**Published:** 2022-06-27

**Authors:** Jennifer L. Cannon, Matthew H. Seabolt, Ruijie Xu, Anna Montmayeur, Soo Hwan Suh, Marta Diez-Valcarce, Filemón Bucardo, Sylvia Becker-Dreps, Jan Vinjé

**Affiliations:** 1Division of Viral Diseases, National Center for Immunization and Respiratory Diseases, Centers for Disease Control and Prevention, Atlanta, GA 30329, USA; rwo8@cdc.gov (R.X.); xda8@cdc.gov (A.M.); qmw0@cdc.gov (S.H.S.); tng7@cdc.gov (M.D.-V.); ahx8@cdc.gov (J.V.); 2CDC Foundation, Atlanta, GA 30329, USA; 3Office of Advanced Molecular Detection, National Center for Emerging & Zoonotic Infectious Diseases, Centers for Disease Control and Prevention, Atlanta, GA 30329, USA; ngr8@cdc.gov; 4Leidos Inc., Reston, VA 20190, USA; 5Ministry of Food and Drug Safety, Cheonju-Si 28159, Korea; 6Center for Infectious Diseases Research, National Autonomous University of Nicaragua—León (UNAN-León), León 21000, Nicaragua; fili_bucardo@hotmail.com; 7Department of Family Medicine and Epidemiology, University of North Carolina Chapel Hill, Chapel Hill, NC 27599, USA; sbd@email.unc.edu

**Keywords:** norovirus, gut microbiome, metagenomics, acute gastroenteritis, longitudinal, Nicaragua, birth cohort, functional genes, probiotic bacteria, vaccine

## Abstract

Noroviruses are associated with one fifth of diarrheal illnesses globally and are not yet preventable with vaccines. Little is known about the effects of norovirus infection on infant gut microbiome health, which has a demonstrated role in protecting hosts from pathogens and a possible role in oral vaccine performance. In this study, we characterized infant gut microbiome changes occurring with norovirus-associated acute gastroenteritis (AGE) and the extent of recovery. Metagenomic sequencing was performed on the stools of five infants participating in a longitudinal birth cohort study conducted in León, Nicaragua. Taxonomic and functional diversities of gut microbiomes were profiled at time points before, during, and after norovirus infection. Initially, the gut microbiomes resembled those of breastfeeding infants, rich in probiotic species. When disturbed by AGE, Gammaproteobacteria dominated, particularly *Pseudomonas* species. Alpha diversity increased but the genes involved in carbohydrate metabolism and glycan biosynthesis decreased. After the symptoms subsided, the gut microbiomes rebounded with their taxonomic and functional communities resembling those of the pre-infection microbiomes. In this study, during disruptive norovirus-associated AGE, the gut microbiome was temporarily altered, returning to a pre-infection composition a median of 58 days later. Our study provides new insights for developing probiotic treatments and furthering our understanding of the role that episodes of AGE have in shaping the infant gut microbiome, their long-term outcomes, and implications for oral vaccine effectiveness.

## 1. Introduction

Acute gastroenteritis (AGE), marked by episodes of diarrhea and/or vomiting, is a common ailment of children, especially during infancy and early childhood. Rotaviruses and noroviruses are the most common cause of AGE among children globally [1]. These viruses are spread primarily by person-to-person transmission or aerosolized vomitus, but also through the ingestion of contaminated food or water, particularly for norovirus [1]. The severity of illness can range from a moderate inconvenience to severe dehydration and can lead to complications requiring emergency room visits or hospitalization and occasionally leading to death [1,2]. The post-licensure era of the rotavirus vaccine has been marked by sharp declines in rotavirus hospitalizations, emergency department visits, and outpatient care [3]. Yet the number of norovirus cases in children has not decreased as there is currently no vaccine or specific treatment (such as an antiviral or monoclonal treatment) available, although vaccine candidates are in Phase II clinical trials [4]. One in six children in the United States under the age of 5 will receive outpatient care for norovirus AGE, 1 in 14 will visit the emergency department, and approximately 1 in 278 will be hospitalized [5]. Globally, noroviruses cause an estimated 685 million episodes and 210,000 deaths each year [6,7]. Probiotics may prevent norovirus AGE or promote a speedy recovery [8], but their mode of action is not well understood.

Gut microbiome development occurs primarily during the first two years of life and is integral to the development of the infant’s immune system [9,10]. Gut microbiome imbalance is associated with malnutrition, obesity, antibiotic treatment, and inflammatory diseases of the gut [11,12]. The ability to recover from AGE may also depend on having a healthy gut microbiome that is compositionally resilient and functionally redundant [13]. Following the infection by two enteric bacteria of AGE (*Vibrio cholera* and enterotoxigenic *Esherichia coli* (ETEC)), the gut microbiomes of children became highly disrupted, but recovery occurred in a predictable pattern [14]. Antibiotic treatment can cause a highly disrupted gut microbiome and a period of recovery, the speed of which is unique to individuals [15]. However, repeated and sequential treatment with antibiotics can lengthen the time it takes for the microbiome to recover and in some cases, the microbiome does not make a complete recovery [16]. Microbiome disturbance following norovirus infection has been recently studied [17,18,19,20,21,22,23]. One study involving adults used a metagenomic approach to investigate longitudinal gut microbiome changes occurring before, during, and after norovirus infection in the participants involved in a human challenge study with the GI.1 strain [21]. No comparable studies with high resolution longitudinal methods have yet been performed in children. Indeed, a key limitation to prior research in this area has been the use of marker gene approaches (i.e., 16S rRNA) as a primary methodology, which lack the resolution needed to more fully characterize the taxonomic and functional communities of the gut microbiome.

In this study, we characterized gut microbiome disturbances and recovery following AGE associated with norovirus among several children under 18 months of age, whose first episode of AGE included norovirus. Uniquely drawing from a longitudinal birth cohort study conducted in León, Nicaragua, we used a shotgun metagenomic approach to characterize the taxonomic and functional microbiome communities present in stool specimens collected at time points before, during, and after norovirus infection and AGE. Our findings will further our understanding of the gut microbiome communities that are altered by norovirus during AGE and those that are associated with recovery, providing foundational data with an eye towards the future development of preventive treatments such as probiotics and vaccines.

## 2. Materials and Methods

### 2.1. Stool Collection and Pathogen Testing

Stools from participants were collected for this pilot sub-study, nested within an ongoing larger longitudinal birth cohort study, the Sapovirus-Associated GastroEnteritis (SAGE) study, conducted in Leon, Nicaragua [24]. The study was approved by the Institutional Review Boards of the National Autonomous University of Nicaragua, León (UNAN-León, Acta Number 45, 2017) and the University of North Carolina at Chapel Hill (Study Number: 16-2079), and CDC (#0900f3eb81c526a7). Infant and household characteristics (i.e., gender, delivery mode, age at sample collection, breastfeeding status, age at introduction of supplemental feeding, antibiotic use, household water, sanitation, and floor type) were collected by study investigators at project initiation and during field visits (Appendix A). Diarrhea was defined as at least 3 stools or a substantial change in stool consistency within a 24 h period. Data were collected on indicators of illness severity including the frequency and duration of diarrhea and/or vomiting, fever, and the presence of rehydration therapy. In the event of AGE, stools were collected by field workers within 2 h of defecation. AGE stools were collected in response to a report of AGE assessed during weekly household visits. Routine stools were collected monthly (once every 4 weekly household visits). Stools were collected in sterile plastic containers or soiled diapers, stored in refrigerators, then were shipped on cold packs to the Centers for Disease Control (Atlanta, GA, USA). The stools were tested for the presence of astrovirus, norovirus (GI/GII), rotavirus, and sapovirus using real-time quantitative polymerase chain reaction (RT-qPCR) [25,26,27]. All AGE stools were additionally tested for the presence of 22 gastrointestinal pathogens by the BioFire^®^ FilmArray^®^ Gastrointestinal (GI) Panel (BioFire Diagnostics), including *Campylobacter* (*jejuni*, *coli* and *upsaliensis*), *Clostridium difficile* (toxin A/B), *Plesiomonas shigelloides*, *Salmonella*, *Yersinia enterocolitica*, *Vibrio* (*parahaemolyticus*, *vulnificus* and *cholerae*), *Vibrio cholerae*, Enteroaggregative *E*. *coli* (EAEC), Enteropathogenic *E*. *coli* (EPEC), Enterotoxigenic *E*. *coli* (ETEC) lt/st, Shiga-like toxin-producing *E*. *coli* (STEC) stx1/stx2, *E*. *coli* O157, *Shigella*/Enteroinvasive *E*. *coli* (EIEC), *Cryptosporidium*, *Cyclospora cayetanensis*, *Entamoeba histolytica*, *Giardia lamblia*, adenovirus F 40/41, astrovirus, norovirus GI/GII, rotavirus A, and sapovirus (I, II, IV and V).

### 2.2. Library Preparation, Metagenomic Sequencing and Quality Control Analysis

The DNA from stool samples was extracted using the QIAamp PowerFecal Pro DNA kit (Qiagen, Hilden, Germany). Briefly, DNA was extracted from 250 mg (solid) or 250 uL (liquid) stool and eluted in 50 µL C6 elution buffer (Qiagen). Nuclease-free water and a defined-composition genomic DNA standard (ZymoBIOMICS Microbial Community Standard, Zymo Research, Irvine, CA, USA) served as the negative and positive controls for the sequencing procedure, respectively. Prior to the library preparation, the concentration of the DNA was measured using the Qubit double stranded DNA high-sensitivity assay (Invitrogen Qubit dsDNA HS assay kit, Invitrogen, Waltham, MA, USA) and was then normalized to 100 ng. Metagenomic libraries were prepared using the Illumina DNA prep kit (Illumina, San Diego, CA, USA) according to the manufacturer’s instructions. The quality and concentration of the prepared library was determined using a 4200 TapeStation DNA Screen device with High Sensitivity D1000 Screen tape (Agilent, Santa Clara, CA, USA), a Qubit HS DNA assay, and a Qubit 2.0 fluorometer (Thermo Fisher Scientific, Waltham, MA, USA). An equimolar mixture (~50 nM each) of the libraries was sequenced as recommended by the manufacturer on an Illumina NovaSeq 6000 instrument for 500 cycles (2 × 250-bp paired-end run). Library demultiplexing and adapter trimming were carried out on the instrument.

Obtained sequencing reads were preprocessed using FastQC to assess data quality (https://www.bioinformatics.babraham.ac.uk/projects/fastqc/, accessed on 25 May 2021). Low-quality bases with a minimum Phred quality score of Q20 were removed using BBDuk (part of the BBTools software package) [28]. Nonpareil3′s kmer mode was used to estimate the fraction of the microbiome community that had been sampled by sequencing, with the aim of achieving 95% coverage or greater [29].

### 2.3. Analysis of Taxonomic Community Composition and Differential Abundance

Community composition of each metagenome was determined using the Kraken2 classification engine and the MaxiKraken database [30] using the parameters “–confidence 0.99” and with all other settings left as default. Read counts per taxon were parsed using custom Perl scripts to derive a merged operational taxonomic unit (OTU) abundance matrix, where each row represented one of the sequenced metagenomes, each column represented one OTU, and the value in the cell contained the OTU’s raw abundance per metagenome. Estimated species richness and Shannon entropy H’ for each metagenome were computed using custom Perl scripts. More granular changes in taxonomic diversity were estimated using the program Mash [31] to compute a distance approximation of the Jaccard index. To establish the baseline levels of microbiome diversity for each child, the first sample that was collected was randomly divided in half to facilitate a ‘self-vs-self’ comparison. All subsequent samples per patient were compared to the sample collected at the preceding time point using Mash distances.

Differentially abundant taxa were identified at the genus level using the package DESeq2 v. 1.30.1 in R [32]. The number of reads classified under each OTU were aggregated together at the species and genus level using the phyloseq v.1.34.0 R package [33]. Reads classified under each unique taxon were then normalized using the “poscounts” method implemented in the DESeq2 R package [32]. Taxa which were differentially abundant between infection phases were identified by fitting the normalized read counts to a Negative Binomial distribution using the “local” fit model, and a likelihood ratio test was performed to determine the best-fitting null model for hypothesis testing. Raw p-values were adjusted using the Benjamini and Hochberg method (*p*-adj) to correct for multiple testing and we defined significance as *p*-adj ≤ 0.05 or *p*-adj ≤ 0.01. Differentially abundant taxa identified from different infection phases were visualized in R using the ggplot2 package [34].

### 2.4. Analysis of Functional Community Composition and Differential Abundance

Metagenomes were assembled using the Nextflow Core MAG pipeline (URL: nf-co.re/mag) [35,36,37]. Open reading frames (ORFs) were identified from each sample’s meta-assembly using Prodigal v. 2.63 [38] with the “normal” mode, which is the mode that identifies ORFs based on the start codons, end codons, and other properties that can be directly learned from the input sequences. To estimate a non-redundant pseudo-“pangenome” of ORFs, CD-HIT v.4.6 [39] was then used to cluster all identified ORFs based on their percentage nucleotide identity using the default threshold of 90% identity. Cluster-representative sequences were extracted and annotated to assign KEGG functional terms (Kyoto Encyclopedia of Genes and Genomes) using kofamscan v. 1.3.0 [40,41] with KOfam’s prokaryotic profile (--profile prokaryote.hal). Briefly, metabolic pathways in the KEGG database are hierarchical structured into three tiers, with descending specificity from Tier 1 to Tier 3. ORFs without KEGG assignment were filtered out and were not considered further in downstream analyses. The functional community of each metagenome was estimated by mapping short reads from each metagenome to the annotated set of representative ORFs using BBMap v. 38.90 [42] and tabulating the number of mapped reads per ORF using HTSeq v. 0.11.2 [43].

Reads mapped to genes involved in the same KEGG pathways were aggregated together to identify differentially abundant (DA) functional pathways across different infection phases with DESeq2 v. 1.30.1 [32], using the same parameters and models described in the taxonomic classification methods. At this stage, data from participant A and the second norovirus infection from participant B were excluded from further differential abundance analyses as there was no evidence of gut microbiome disturbance during these episodes of AGE. This was done to streamline direct comparisons between the disturbed microbiomes (i.e., during AGE) and the microbiomes sampled before and after the disturbance. Since our primary focus was on the functions associated with microbial metabolic activities, the KEGG pathways associated with (Tier 1) “Human Diseases”, “Organismal Systems”, and “Brite Hierarchies” were excluded from further analysis. The relative abundances of each KEGG pathway were calculated as the ratio of reads mapped to a given pathway divided by all reads mapped to the pathways in the same KEGG tier. Paired one-sided t-tests were used to determine the Tier 2 pathways that were significantly different in relative abundance between the before versus during and the before versus after AGE samples (*p*-adj < 0.01). The relative abundances of the Tier 2 pathways were then further broken down to examine the relative abundance of the Tier 3 pathways umbrellaed underneath by recalculating the relative abundance of each Tier 3 pathway using the ratio described above.

## 3. Results

### 3.1. Metadata Summary

We selected the five of the nine children enrolled in the SAGE birth cohort who experienced norovirus-associated gastroenteritis as their first episode of AGE in life and who met our inclusion criteria for age and norovirus strain (Table 1). Three children were excluded because, unlike the five included children, they were not infected with a GII.4 Sydney strain, and one was excluded because the child was less than 6 months of age during the AGE episode. All the stool samples for the study were collected between October 2018 and September 2019. Prior to this event, none of the stools collected from these children during routine monthly visits tested positive using the enteric virus panel (norovirus, sapovirus, rotavirus, or astrovirus) and no mixed viral infections were detected. Two of the children were co-infected with EPEC (enteropathogenic *E. coli*) during the AGE episode and one was additionally co-infected with EAEC (enteroaggregative *E. coli*) (Table 1). Four of the five children (participants A, B, D, E) received two doses of Rotarix, while participant C received one dose of Rotarix. The five children ranged in age from 8 to 14 months at the time of their first episode of norovirus-associated AGE, which was typed as GII.4 Sydney[P16] (Table 1). All the children were male, with 60% (3/5) being delivered vaginally and 40% by Cesarean. All children were breastfed throughout the sample collection dates and supplementary foods or drinks were part of the children’s normal diet. All but one of the mothers (mother of participant C) and all children were ‘secretors’, capable of expressing histo-blood group antigens (HBGA) on the surface of gut mucosal cells and their secretions. Household characteristics were also similar for all children (Appendix A). The severity of illness assessed by a modified Vesikari score [24] ranged from 6 to 10 on a 15-point scale (Table 1). One participant (B) was treated with antibiotics prior to or during the first AGE episode (Figure 1). This child experienced a subsequent GII norovirus infection (GII.14[P7]) 8 months after the initial GII.4 Sydney[P16] infection. During this norovirus episode, the child was coinfected with rotavirus, astrovirus, and ETEC (enterotoxogenic *E. coli*). Our intent was to analyze stools collected just before norovirus-associated AGE, during AGE, and those collected 1 and 2 months after the illness. With the longitudinal design, the samples collected before AGE served as negative controls for each child in lieu of matched controls from the greater study. Choosing from the available stool samples, metagenomic analysis included norovirus-positive samples collected during the AGE episodes, stools collected during routine household visits 7 to 93 days (median = 16 days) prior to episodes of AGE, and stools collected during routine visits 12–114 days (median = 58 days) after norovirus was detected (Figure 1). Since participant B experienced a second episode of GII norovirus-associated AGE, additional stool samples were tested for participant B before, during, and after the second GII norovirus infection.

### 3.2. Metagenome Sequencing Coverage

Metagenome coverage, defined as the fraction of the microbial community captured during sequencing, was assessed by a Nonpareil analysis (Appendix A) [29]. For all samples regardless of the phase of norovirus infection, coverage exceed the desired 95% minimum threshold, satisfying that statistically robust conclusions could be drawn from the metagenomic data. Rarefaction curves also show that samples were sufficiently sequenced, as the number of unique species detected in each metagenome increased initially with a greater read depth but began to plateau for many samples by 3 million reads, and for all samples by the 5 million read depth (Appendix A).

### 3.3. Taxonomic Diversity before, during and after Norovirus-Associated AGE

Nearly all the classified reads (95–99% per sample) were classified as bacteria; Archaea, Eukarya (mostly yeasts), and viruses together comprised less than 1% of the classified reads per sample (Appendix A). The metagenomes from stool samples collected prior to a child’s first episode of norovirus-associated AGE (samples S1, S4, S11, S15, and S19) were rich in Actinobacteria (65–92% of classified reads) (Figure 2a). Bacilli, Coriobacteria, and Clostridia classes together comprised less than 20% of classified reads in these ‘before’ infection samples. In contrast, metagenome compositions of samples collected during norovirus-associated AGE were richly abundant with Gammaproteobacteria for four of the five children (B, C, D, and E). A range of 49 to 87% of reads were classified as such. Metagenome composition shifted again after AGE, which was shown in the stool samples collected 12 to 114 days after norovirus infection. A reversion to dominance by Actinobacteria was observed, with Clostridia, Bacilli, and Coriobacteria comprising the next most abundant proportion of classified reads. Alpha taxonomic diversity (measured using Shannon’s H’) increased during infection for these four children but returned to similar levels after AGE symptoms resolved (Figure 2b). The Inverse Simpson’s (1/D) index showed a similar pattern of change in alpha diversity (Appendix A; Appendix A). Species richness measured by the Chao1 index showed a slightly different pattern, with the number of species present after AGE being generally greater than the number detected prior to norovirus infection (Appendix A; Appendix A). For one child (Participant A), the gut microbiome did not change during the AGE episode despite a coinfection with EPEC and EAEC. We observed that Actinobacteria remained the class with the highest abundance for this child before, during, and after norovirus infection, with no notable enrichment of Gammaproteobacteria during the AGE episode. Participant B experienced a second GII norovirus infection 38 weeks after the initial GII.4 Sydney illness ended (Figure 1). For this child, although coinfection with ETEC, rotavirus, and astrovirus was detected, the gut microbiome composition did not change with the second infection (Figure 2a). However, there was a shift in the microbiome composition noticed 37 weeks after the first episode of AGE ended, prior to the second GII norovirus infection. This shift included a higher proportion of Bacilli and Clostridia observed when compared to previous samples collected after the first episode of norovirus-associated AGE (Figure 2a). Alpha diversity levels did not change for this child during the second norovirus infection but did increase prior to the child’s second GII norovirus infection (Figure 2b).

To quantify changes in diversity between samples longitudinally, we compared Mash distances in a pairwise fashion between sampling times per child (Figure 3). Following the establishment of baseline (pre-infection) diversity, Mash distance values increased during infection and returned to nearly baseline levels in the weeks after infection. This was the case also for Participant A, although the shift was less pronounced than in the other children. The metagenome composition also shifted before the second GII norovirus infection for Participant B, did not change during infection, and increased again after infection.

### 3.4. Species Abundance before, during and after Norovirus-Associated AGE

For the children whose microbiomes were disrupted during norovirus-associated AGE, 316 taxa at the genus level were differentially abundant from taxa present before infection at the *p*-adj < 0.05 level (Appendix A). The number of differentially abundant genera by Class are shown in Table 2. Those that demonstrated a >5 log2-fold change at the *p*-adj < 0.01 level are shown in Figure 4 for visual clarity. Clostridia (35 genera, including Blautia, Roseburia, and Ruminococcus), Bacilli (26 genera, including Lactococcus, Weissella, and Gemella), and Actinobacteria (31 genera, including Actinomyces and Bifidobacterium) were significantly more abundant before infection. Four Bifidobacterium (*B. longum*, *B. breve*, *B. bifidum*, and *B. kashiwanohense*) ranked among the top three species in each child’s microbiome prior to infection, together comprising 78 to 89% of all bacterial species (Appendix A).

### 3.5. Changes to Gut Microbiome Metabolic Function during Norovirus-Associated AGE

There were 3499 differentially abundant genes identified when comparing the gut microbiomes before versus during disruptive norovirus-associated AGE at the *p*-adj < 0.05 level of significance (Appendix A). Of these, 468 genes exhibited a >10 log2-fold increase in abundance at the *p*-adj < 0.01 level of significance during microbiome disruption (Appendix A). These genes included those that are primarily involved in polysaccharide and lipopeptide biosynthesis, which are needed for cell wall formation, including the lipopolysaccharide (LPS) layer of gram-negative bacteria, as well as two-component system kinases used in sensing and responding to environmental stimuli. For visual clarity, the top 50 differentially abundant genes with the highest log2-fold change values during disturbance are shown in Figure 5. Other genes that were in greater abundance during disruptive AGE included multidrug efflux pump components, pilus assembly, and conjugal transfer proteins, which could have a role in bacterial defense and antibiotic resistance. Among the 276 genes that were more abundant (>5 log2-fold absolute value change at the *p*-adj < 0.01 level of significance) prior to norovirus-associated AGE than during infection were genes involved in the cleavage and transfer of sugar groups (for example, fucosidase, fucosyltransferase, fucose synthase, and fucose mutarotase) (Appendix A).

Looking more broadly at the changes in metabolic function associated with disruptive norovirus-associated AGE, we found 17 Tier 2 KEGG pathways that differed significantly (*p*-adj < 0.01) when comparing the microbiomes of children before infection with the disrupted microbiomes of children during AGE (Appendix A; Figure 6). The relative abundance of Tier 2 KEGG pathways present in the microbiomes of participant A and the second infection of participant B during AGE was distinct from the disturbed microbiomes; they resembled more closely the before and after AGE microbiomes of all children (Figure 6). Pathway detections that significantly decreased in abundance in the disrupted microbiomes included carbohydrate metabolism, glycan biosynthesis and metabolism, and replication and repair. Those that increased with disruption included amino acid metabolism, signal transduction, and cellular community–prokaryote processes. Taking a closer look at the pathways involved in carbohydrate metabolism or glycan biosynthesis, Tier 3 KEGG pathways involved in carbohydrate metabolism that were fewer in abundance in the disrupted microbiomes included those involved in the metabolism of starch, sucrose, galactose, and those involved in the degradation of ‘other’ glycans (Figure 7a). KEGG tier 3 pathway detections included those involved in glycan biosynthesis that were higher in abundance during disruption, including those involved in O-Antigen nucleotide sugar and peptidoglycan biosynthesis (Figure 7b).

When comparing the functional gene complement of microbiomes before and after disruptive norovirus-associated AGE, only 20 differentially abundant genes were found (*p*-adj < 0.05), with a maximum log2-fold change of 3.6 (Appendix A). Similarly, no Tier 2 KEGG pathways were differentially abundant when comparing before vs. after infection microbiomes (*p*-adj < 0.05).

## 4. Discussion

We longitudinally studied the gut microbiomes of five children enrolled in a Nicaraguan birth cohort who, within the first 14 months of life, experienced a norovirus infection associated with their first episode of AGE. Our results showed similar trends among these microbiomes with regards to their taxonomic and functional diversities prior to, during, and following disturbances due to AGE, and most notably, our findings show that norovirus-associated AGE did not alter the trajectory of healthy microbiome development for these children during these critical early years.

Detecting Gammaproteobacteria species and enteric pathogens in disturbed gut microbiomes is not unique to norovirus-associated AGE, but rather a generalized response to AGE [14]. Mucosal cell damage, loss of barrier function, and an influx of oxygen to the intestinal lumen are consequences of diarrhea. Obligate anaerobes are succeeded by aerobic and facultative anaerobic bacteria that are typically present in a low abundance in the healthy gut [14,17,19]. Two of the children in our study were coinfected with EPEC or EPEC/EAEC during their first norovirus-associated AGE episode. Although these *E. coli* strains are associated with diarrhea, it is also common for them to be detected in asymptomatic stools and in coinfections [44,45]. These two children also experienced vomiting, which is typical of norovirus AGE but not a symptom of EAEC or EPEC [44], suggesting, but not confirming, that norovirus was the primary agent of AGE in these children. *Pseudomonas* spp. were highly abundant in the disturbed microbiome of the infants in our study. Typically, they are inhabitants of soil and aquatic environments and sometimes are associated with milk and food spoilage [46,47]. They are low-abundance commensal bacteria of the gut and breastmilk and apart from *Pseudomonas aeruginosa*, they rarely cause disease in humans. The *Pseudomonas* selection and outgrowth seen in the disturbed microbiomes in our study might relate to the selective pressure of the oxygenated environment due to diarrhea, their proficiency in sequestering iron, and their competitive bacterial defense mechanisms. Supporting this hypothesis, signal transduction genes, including those involved in osmoregulation, environmental sensing, and defense, were higher in abundance during microbiome disturbance. One example is the siderophore enterobactin; *Pseudomonas* spp. are adept at using siderophores for sequestering iron from the environment, making it less available to other bacteria as well as host cells [48]. It remains unclear if the increased abundance of *Pseudomonas* spp. during disruptive AGE is a generalized response for breastfeeding infants, infants in low-income settings, or a community-specific response of the Nicaraguan children in the study.

Following norovirus-associated AGE, there was a return of the probiotic species (i.e., *Bifidobacterium* spp. and *Lactobacillus* spp.) and functional genes that were highly abundant prior to AGE (such as those responsible for the metabolism of carbohydrates and lipids in breastmilk), indicating that the healthy infant gut microbiome was restored. One participant (Participant A) did not experience gut microbiome disturbance during norovirus-associated AGE despite having a moderate illness (severity score of 10) and a coinfection with EAEC/EPEC. Instead, microbiome diversity decreased during infection and nearly all the species detected were *B. longum* (96% relative abundance). It has been reported previously that not all children and adults experience gut microbiome disturbance during norovirus-associated AGE [17,19,20,23] for reasons that are still unknown. Another child experienced two symptomatic GII norovirus infections within 9 months, the second one being a coinfection of GII.14[P7] norovirus, rotavirus, adenovirus, and ETEC, but this infection did not disrupt the gut microbiome. This child vomited only one time and did not have diarrhea, suggesting the weakened severity of illness was due to cross-protective adaptive immunity or innate immune defenses rather than gut microbiome community or function. This child also received multiple doses of antibiotics before and after the two norovirus episodes when AGE symptoms were not present, but without a loss of diversity or gut microbiome changes that are commonly associated with antibiotic use. Since the sequencing depth was sufficient to be able to characterize gut microbiome changes, it is possible that breastfeeding or other nutritional or environmental factors countered the effect of antibiotic use for this child. This finding was unexpected and warrants further investigation.

Norovirus-associated AGE did not alter the trajectory of gut microbiome development for infants enrolled in a birth cohort study in Nicaragua. The consequences of AGE on gut microbiome structure and function were not long-term or severe in these children. Larger and more diverse study populations are needed to substantiate our findings and explore the alternative outcomes that likely exist within and between populations. We acknowledge the small sample size used in this study to be the primary limitation for interpreting our results in a broader context; thus, we consider this work to be a pilot for future studies. Nevertheless, an important aspect of our study was to observe gut microbiome changes within participants that were immunologically naïve and whose microbiomes had not yet been shaped by disruption due to AGE. Additionally, microbiome composition might be associated with an ability to mount a robust immune response following rotavirus vaccination, although supportive data are conflicting [49]. This might also be the case for future norovirus vaccines and warrants further study. The large amount of metagenomic data collected in this study can be leveraged and extended. For example, a follow-up study could include children that did not experience a norovirus (or other enteric virus) infection during the first year of life, attempting to identify potentially probiotic species that are highly abundant in the gut microbiomes of these children when compared to those having an episode(s) of norovirus-associated AGE. Long-term health consequences and gut microbiome changes might also be illuminated following repeated episodes of AGE or asymptomatic shedding, as has been recently studied for *Campylobacter* [50]. Including norovirus-negative diarrheal stools would also help differentiate gut microbiome changes due to norovirus AGE versus those from all-cause diarrhea. Characterizing the gut microbiomes of infants and young children in health, with norovirus AGE, and after recovery thus has implications for therapeutic and preventive treatments, especially in the development of probiotics and vaccines.

## Figures and Tables

**Figure 1 viruses-14-01395-f001:**
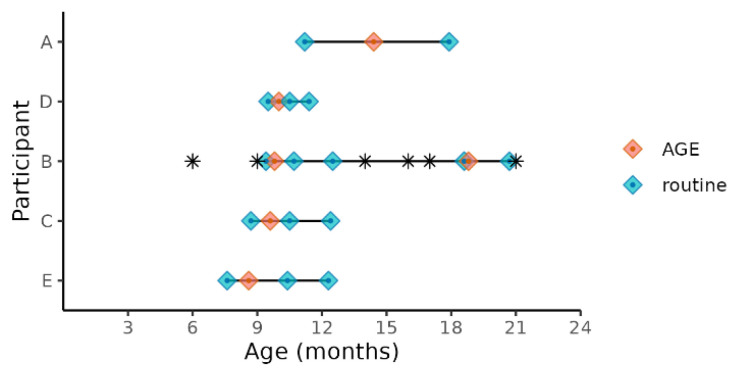
Child age at sample collections times before, during, and after norovirus-associated AGE. Diamonds indicate the child’s age at sample collection time points. Samples collected before and after AGE are indicated as “routine”. The children’s ages when antibiotics were administered are indicated by *****.

**Figure 2 viruses-14-01395-f002:**
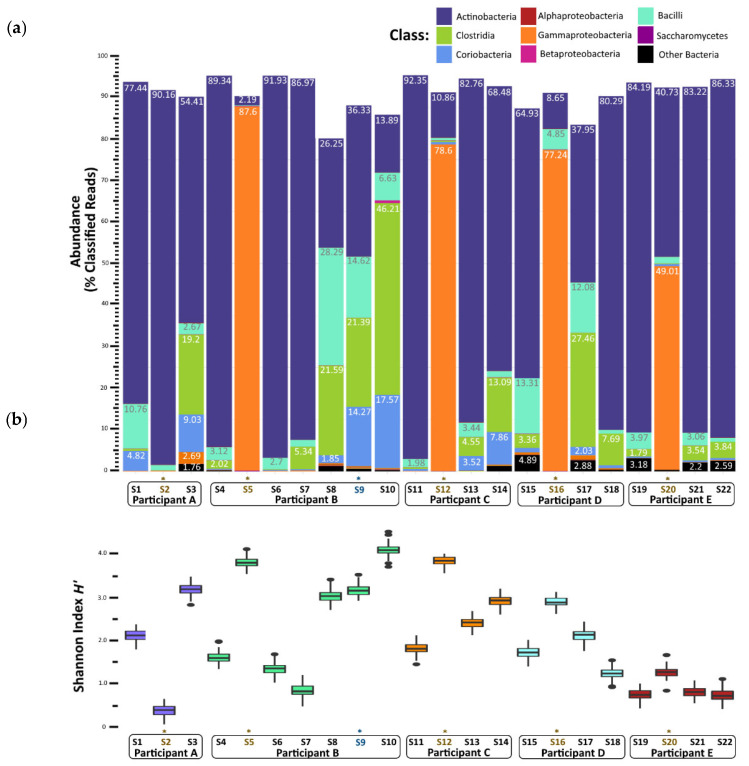
Gut microbiome changes in composition and diversity before, during, and after norovirus-associated AGE. (**a**) Percentage abundance changes at the Class rank. (**b**) Alpha diversity measured by the Shannon entropy H’ Index. An * and brown colored text represent the samples collected during the AGE episodes. For Patient B, a second norovirus infection occurred, indicated by the blue text. Numbers following “S” indicate the sample identifier for metagenomic data.

**Figure 3 viruses-14-01395-f003:**
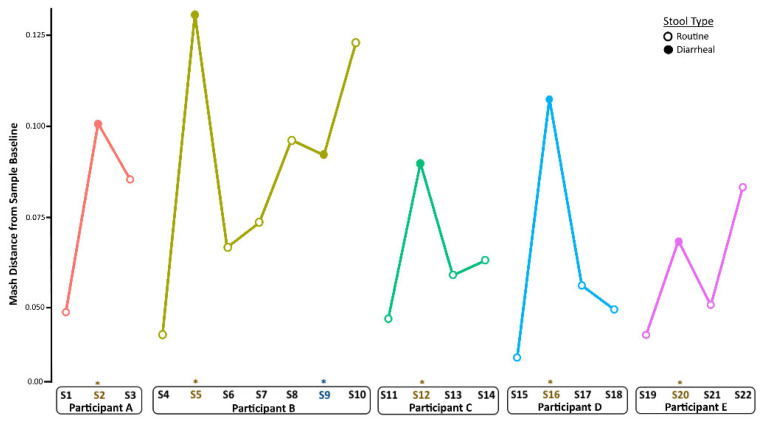
Longitudinal changes in taxonomic diversity measured by Mash distance. Metagenome composition increased from baseline (pre-infection) levels during norovirus infection and decreased in the weeks after recovery from AGE for all children. An * and brown colored text represent the samples collected during the AGE episodes. For Patient B, a second norovirus infection occurred, indicated by the blue text. Numbers following “S” indicate the sample identifier for metagenomic data.

**Figure 4 viruses-14-01395-f004:**
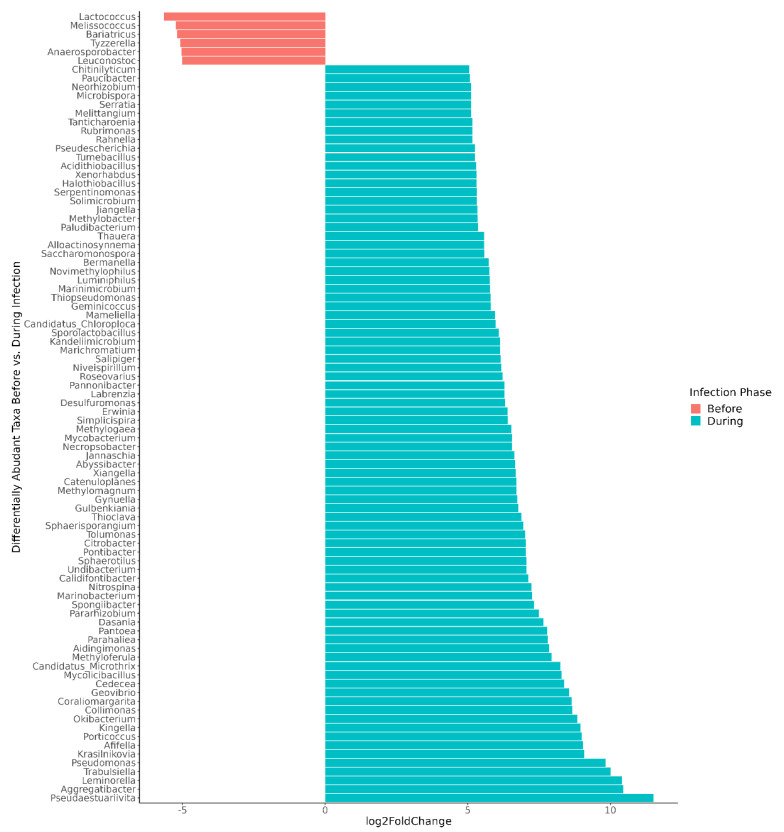
Differentially abundant genera, comparing microbiomes before versus during norovirus-associated AGE. Taxa demonstrating a >5 log2-fold change and adjusted *p*-values < 0.01 are shown in the figure. In all, 316 differentially abundant genera were found at the *p*-adj < 0.05 level. Participant A was excluded from the genus-level differential abundance analysis since the gut microbiome was not disrupted during AGE.

**Figure 5 viruses-14-01395-f005:**
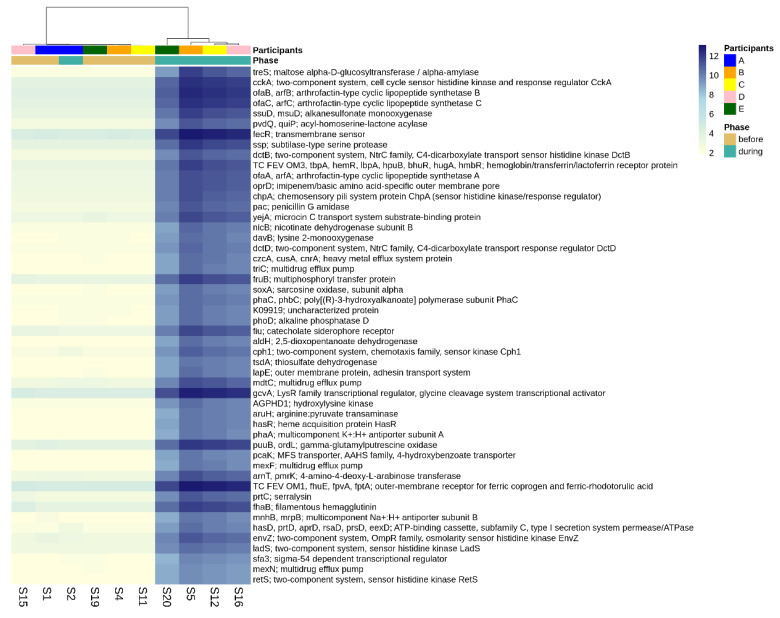
Differential abundance of functional genes comparing pre-infection microbiomes to those disturbed by norovirus-associated AGE. The first-row color scale indicates the participant numbers. The second row shows the phase of sample collection; before or during AGE. Numbers following “S” indicate the sample identifier for metagenomic data. Figure shows the top 50 differentially abundant genes with highest log2-fold change during disturbance (*p*-adj < 0.05). There was a total of 3499 differentially abundant KO genes detected at the *p*-adj < 0.05 level.

**Figure 6 viruses-14-01395-f006:**
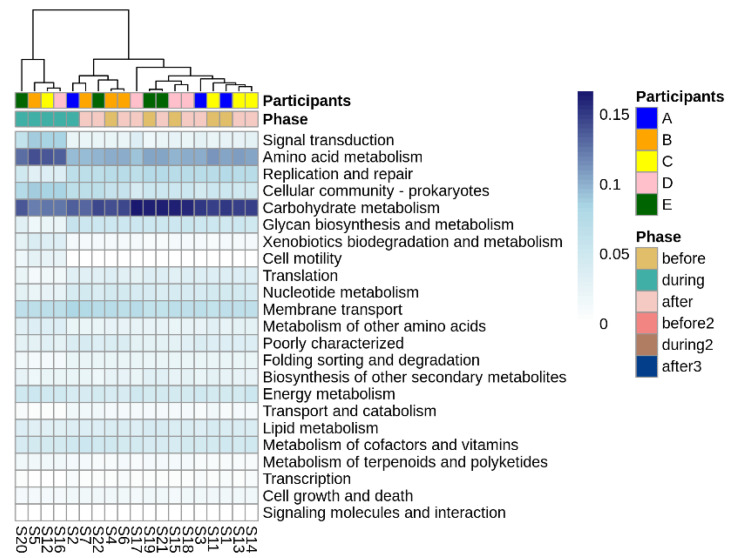
Differentially abundant Tier 2 KEGG pathways before, during, and after disruptive norovirus-associated AGE. The first-row color scale indicates the patient numbers. The second row shows the phase of sample collection; before, during, or after AGE. Numbers following “S” indicate the sample identifier for metagenomic data. Note that for Participant A and for the second norovirus infection of Participant B, the during infection microbiomes are grouped with the ‘before’ and ‘after’ infection the microbiomes of all children, including their own.

**Figure 7 viruses-14-01395-f007:**
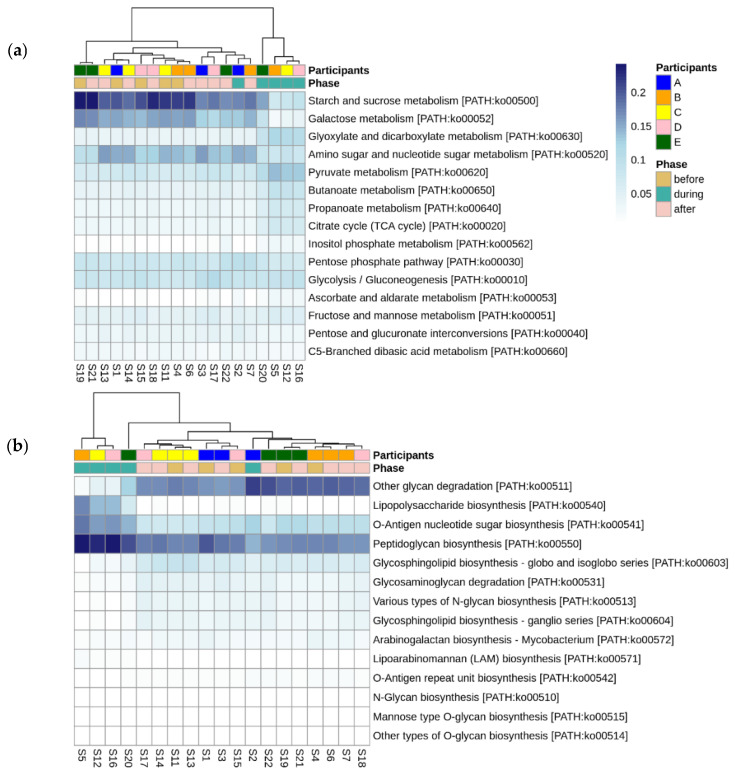
Differentially abundant Tier 3 KEGG pathways before, during, and after disruptive norovirus-associated AGE. Those within the Tier 2 KEGG pathways of (**a**) Carbohydrate metabolism, and (**b**) Glycan biosynthesis are shown. The first-row color scale indicates the participant numbers. The second row shows the phase of sample collection; before, during, or after AGE. Numbers following “S” indicate the sample identifier for metagenomic data. Note that for Participant A and for the second norovirus infection of Participant B, the microbiomes during norovirus-associated AGE are grouped with the ‘before’ and ‘after’ infection microbiomes of all children, including their own.

**Table 1 viruses-14-01395-t001:** Characteristics of the children, their birth, and their first episode of norovirus-associated acute gastroenteritis (AGE) for participants in the Nicaraguan birth cohort submitting stools for microbiome analysis.

Participant	Gender	Delivery Mode	Child Age at First Case of AGE (Months)	Norovirus Strain	Bacterial/Viral Co-Infections ^1^	Severity of Illness Score ^2^
A	Male	Vaginal	14.3	GII.4 Sydney[P16]	EAEC, EPEC	10
B	Male	Cesarean	9.8	GII.4 Sydney[P16]	None	9
C	Male	Vaginal	9.5	GII.4 Sydney[P16]	None	6
D	Male	Vaginal	9.9	GII.4 Sydney[P16]	EPEC	9
E	Male	Cesarean	8.4	GII.4 Sydney[P16]	None	8

^1^ EAEC, enteroaggregative *E. coli*; EPEC, enteropathogenic *E. coli*; ^2^ The clinical severity of AGE episodes was described using a scale of 0–15, in which points were assigned based on symptom severity (Diarrhea lasting 1–2 days = 1 point; 3–4 days = 2 points; 5+ days = 3 points. Vomiting lasting 1–2 days = 1 point; 3–4 days = 2 points; 5+ days = 3 points. Maximum of 4–5 stools per day = 1 point; 6–7 stools = 2 points; 8+ stools = 3 points. Presence of fever = 3 points. Received intravenous fluid for dehydration = 3 points).

**Table 2 viruses-14-01395-t002:** **The** number of unique genera that were differentially abundant ^1^ before AGE versus during norovirus-associated AGE, by bacterial class.

Bacterial Class	Before Infection	During Infection
Clostridia	35	
Bacilli	26	2
Actinobacteria	31	24
Gammaproteobacteria	2	65
Betaproteobacteria		41
Alphaproteobacteria	5	37
Deltaproteobacteria		7
Erysipelotrichia	6	1
Coriobacteria	4	
Cytophagia	2	4
Flavobacteria	2	1
Bacteroidia	2	
Mollicutes	2	
Chlamydia	1	
Acidimicrobia		1
Acidithiobacillia		1
Chloroflexia		1
Deferribacteres		1
Nitrospinia		1
Opitutae		2
Planctomycetia		1
Spirochaetia		1

^1^ Includes all differentially abundant genera, adjusted *p* < 0.05.

## Data Availability

All available metagenomic data are Bioproject accession number PRJNA853174 in the NCBI BioProject database (https://www.ncbi.nlm.nih.gov/bioproject/).

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
