# Peer review of "Gut Microbiome Changes Occurring with Norovirus Infection and Recovery in Infants Enrolled in a Longitudinal Birth Cohort in Leon, Nicaragua"

_viruses, 2022, doi:10.3390/v14071395_

Round 1

Reviewer 1 Report

Congratulations to the authors on a well-designed and executed sub-study, that addresses questions not explicitly envisioned in the design of the original, larger parent study. I agree with the authors’ characterization of this as a “pilot for future studies”, and perhaps that could be given more emphasis given the acknowledged small sample size. Could they perhaps start the methods section with “In this pilot sub-study, nested within an ongoing larger longitudinal birth cohort…”? Could they also explicitly state the reason for the small sample size (presumably the costliness of the metagenomic sequencing)? Another limitation is the lack of a comparison group. This is implicit in lines 436-37 but could perhaps be mentioned explicitly.

Some subheadings in the methods section would help the reader to navigate (e.g., “Microbiological analysis”, “Statistical analysis”).

I recommend some engagement in the discussion with this study https://academic.oup.com/cid/article/71/4/1000/5571186?login=true which is also from Latin America and looked at a similar question but for Campylobacter rather than norovirus. It seems the conclusions for that pathogen contrast with the norovirus reported presented here.

Otherwise, I see no reason to delay the publication of these important findings.

Author Response

Congratulations to the authors on a well-designed and executed sub-study, that addresses questions not explicitly envisioned in the design of the original, larger parent study. I agree with the authors’ characterization of this as a “pilot for future studies”, and perhaps that could be given more emphasis given the acknowledged small sample size. Could they perhaps start the methods section with “In this pilot sub-study, nested within an ongoing larger longitudinal birth cohort…”?

Thank you for your helpful comments in your review. We’ve added text in the first sentence of the Materials and Methods section to emphasize this is a pilot sub-study.

Could they also explicitly state the reason for the small sample size (presumably the costliness of the metagenomic sequencing)?

The reason for the small sample size was, to date, there were only 9 children that had their first AGE episode associated with norovirus. Of these 9, only 5 met the inclusion criteria of the study (all infected with GII.4 Sydney virus and at least 6 m of age). In a larger study, we expect there to be more instances of first AGE episodes norovirus. We now describe the inclusion criteria in the first paragraph of the Results section. And yes- the larger study will be quite expensive.

Another limitation is the lack of a comparison group. This is implicit in lines 436-37 but could perhaps be mentioned explicitly.

We added the following sentence to section 3.1 of the Results section: “With the longitudinal design, the samples collected before AGE served as negative controls for each child in lieu of matched controls from the greater study.”

Some subheadings in the methods section would help the reader to navigate (e.g., “Microbiological analysis”, “Statistical analysis”).

              Yes, this makes it easier to read- done.

I recommend some engagement in the discussion with this study https://academic.oup.com/cid/article/71/4/1000/5571186?login=true which is also from Latin America and looked at a similar question but for Campylobacter rather than norovirus. It seems the conclusions for that pathogen contrast with the norovirus reported presented here.

Thank you for bringing this study to our attention. In the study they found asymptomatic Campylobacter carriage to be associated with changes in the gut microbiome that could lead to growth faltering in a population with high burden enteric diseases and stunting. In this pilot study, we did not follow the children beyond the AGE episodes described. It is possible that repeated AGE episodes or asymptomatic norovirus shedding could lead to long-term changes in gut microbiome diversity or health of the children. We added the following sentence to the Discussion to indicate this would be an interesting follow-up study to perform: “Long-term health consequences and gut microbiome changes might also be illuminated following repeated episodes of AGE or asymptomatic shedding as has been recently studied for Campylobacter”.

Otherwise, I see no reason to delay the publication of these important findings.

Reviewer 2 Report

In this manuscript, The authors performed metagenomics sequencing on samples collected from 5 infants during norovirus-associated AGE and during pre- and post- norovirus-associated AGE. They have observed changes in gut microbiome composition and functionality during norovirus-associated AGE and which were eventually resolved once the infection was resolved. The findings seem very interesting. However, several concerns need to be addressed to improve the quality of the manuscript.--

Major concerns:

1. The authors used qPCR to run a viral panel assay and found that there were no other viral infections other than the norovirus. However, AGE can be caused by different bacteria and parasites as well. Did the authors check for other AGE-causing pathogens? How did the authors confirm that observed AGE was due to norovirus but not other pathogens. It would be great if they could run a qPCR/TaqMan array for particular AGE-causing pathogens, including Campylobacter, E. coli, Salmonella, Shigella, and E.histolytica, Crypto, Giardia.

2. Among six norovirus-positive samples, 5 were diarrheal samples. Diarrheal episodes themselves are sufficient to alter the composition of gut microbiota. How do the authors confirm that the observed changes in gut microbiota were due to symptomatic norovirus infection but not just the diarrheal episodes? It would be helpful to include some norovirus negative diarrheal samples in the analysis to conclude that the observed changes in gut microbiota were due to norovirus-specific AGE but not just the diarrheal episodes.

Minor concerns:

In the method section, authors should separate each section/method using a title/name

Author Response

In this manuscript, The authors performed metagenomics sequencing on samples collected from 5 infants during norovirus-associated AGE and during pre- and post- norovirus-associated AGE. They have observed changes in gut microbiome composition and functionality during norovirus-associated AGE and which were eventually resolved once the infection was resolved. The findings seem very interesting. However, several concerns need to be addressed to improve the quality of the manuscript.--

Major concerns:

The authors used qPCR to run a viral panel assay and found that there were no other viral infections other than the norovirus. However, AGE can be caused by different bacteria and parasites as well. Did the authors check for other AGE-causing pathogens? How did the authors confirm that observed AGE was due to norovirus but not other pathogens. It would be great if they could run a qPCR/TaqMan array for particular AGE-causing pathogens, including Campylobacter, E. coli, Salmonella, Shigella, and E.histolytica, Crypto, Giardia.

Thank you for this suggestion. We went back to the lab and performed analysis for the presence of 22 gastrointestinal pathogens by the BioFire® FilmArray® Gastrointestinal (GI) Panel (added to the Methods section). We found that two children were coinfected with EPEC and EPEC/EAEC during the AGE episodes (one with gut microbiome disruption and the other without). The child with a subsequent GII.14 norovirus infection was coinfected with ETEC, rotavirus and astrovirus, but experienced only vomiting and no microbiome changes. We’ve modified the Results section accordingly.

Since we cannot confirm the causative agent of the AGE episodes for these children, we’ve changed throughout the manuscript where we’ve stated, “norovirus AGE” to “norovirus associated AGE”. We also include the following in the Discussion section, “Although these E. coli strains are associated with diarrhea, it is also common for them to be detected in asymptomatic stools and in coinfections [44, 45]. These two children also experienced vomiting, which is typical of norovirus AGE but not a symptom of EAEC or EPEC [45], suggesting, but not confirming, norovirus was the primary agent of AGE in these children”.

Two references were added in support:

  1. Hu, J. and Torres, A.G. Enteropathogenic Escherichia coli: foe or innocent bystander?. Clinical microbiology and infection: the official publication of the European Society of Clinical Microbiology and Infectious Diseases, 2015, 21, (8), 729–734.
  2. Jesser, K.J. and Levy, K., Updates on defining and detecting diarrheagenic Escherichia coli Current Opinion in Infectious Diseases 2020 33, (5), 372-380.

Among six norovirus-positive samples, 5 were diarrheal samples. Diarrheal episodes themselves are sufficient to alter the composition of gut microbiota. How do the authors confirm that the observed changes in gut microbiota were due to symptomatic norovirus infection but not just the diarrheal episodes? It would be helpful to include some norovirus negative diarrheal samples in the analysis to conclude that the observed changes in gut microbiota were due to norovirus-specific AGE but not just the diarrheal episodes.

As per the previous comment, changing “norovirus AGE” to “norovirus associated AGE” throughout the manuscript helps in clarifying this. Previously, we reflected on this in the second paragraph of the discussion, stating that the gut microbiome changes we observe are also what would be expected of diarrhea in general. We now elaborate that this would be an important set of samples to include in a follow-up study. We’ve added the following sentence to the discussion “Including norovirus-negative diarrheal stools would also help differentiate gut microbiome changes due to norovirus AGE versus those from all-cause diarrhea”.

Minor concerns:

In the method section, authors should separate each section/method using a title/name

              Yes, this makes it easier to read- done.

Round 2

Reviewer 2 Report

Please mention the name of the 22 pathogens in the method section that were tested by the BioFire® FilmArray® Gastrointestinal (GI) Panel.

Author Response

We've now listed the names of the 22 pathogens in the Biofire GI pathogens panel in the Methods section.